# communications
# engineering

# Neuronal activity under transcranial radio-frequency stimulation in metal-free rodent brains in-vivo

Omid Yaghmazadeh [1,4 ✉], Mihály Vöröslakos[1,4], Leeor Alon[2], Giuseppe Carluccio[2], Christopher Collins[2], Daniel K. Sodickson[1,2] & György Buzsáki [1,3 ✉]

As the use of Radio Frequency (RF) technologies increases, the impact of RF radiation on neurological function continues to receive attention. Whether RF radiation can modulate ongoing neuronal activity by non-thermal mechanisms has been debated for decades. However, the interactions between radiated energy and metal-based neural probes during experimentation could impact neural activity, making interpretation of the results difficult. To address this problem, we modified a miniature 1-photon $Ca^{2+}$ imaging device to record interference-free neural activity and compared the results to those acquired using metal-containing silicon probes. We monitored the neuronal activity of awake rodent-brains under RF energy exposure (at 950 MHz) and in sham control paradigms. Spiking activity was reliably affected by RF energy in metal containing systems. However, we did not observe neuronal responses using metal-free optical recordings at induced local electric field strengths up to 230 V/m. Our results suggest that RF exposure higher than levels that are allowed by regulatory limits in real-life scenarios do not affect neuronal activity.

[1] Neuroscience Institute, School of Medicine, New York University, New York, NY 10016, USA. [2] Department of Radiology, School of Medicine, New York University, New York, NY 10016, USA. [3] Department of Neurology, School of Medicine, New York University, New York, NY 10016, USA. [4] These authors contributed equally: Omid Yaghmazadeh, Mihály Vöröslakos. ✉email: Omid.Yaghmazadeh@gmail.com; gyorgy.buzsaki@nyulangone.org

The widespread use of Radio Frequency (RF) electromagnetic fields in various technologies has exposed our environment and its living species to ever-growing doses of this type of nonionizing radiation. As a result, the effects of RF radiation on biological tissue have been the subject of intensive research for many decades[1–3]. Possible effects of RF radiation on the brain (and the nervous system in general) have been the focus of particular attention because of the concern that the electrical properties of neurons might render them especially vulnerable to electromagnetic radiation. The advent of cellular communications and the omnipresence of cell phones has lent new urgency to research on the effects of RF fields on brain tissue[4–8]. Since RF fields are also used in magnetic resonance imaging (MRI), RF effects on the brain have also been explored in the imaging literature[9,10]. Many experimental investigations have been conducted exploring RF effects in tissue culture, in brain slices *in-vitro*, in animal experiments *in-vivo* and in humans, using EEG and cognitive behavior. A conflicting set of changes—or absence of changes—in various parameters has been reported, ranging from alterations in specific gene expression in neurons to blood brain barrier breakage to EEG perturbations to changes in overt behavior and cognition. The findings of these studies vary from no appreciable effects to harmful consequences on brain activity, using a variety of RF frequencies, wide ranges of power and stimulation patterns[4,5,11–33].

An ongoing debate surrounding neural effects of RF in the microwave range involves the possibility, or absence, of a non-thermal mechanism besides RF-induced thermal changes. This debate has continued since the observation of "RF hearing effects" in the early 1960's[5,34,35]. However, there have been few direct studies of the instantaneous (and potentially nonthermal) effects of RF exposure on neuronal activity[13,14,26,27,36–39]. A particular challenge for such studies is to identify a technique for monitoring neuronal activity that is not affected by RF interference. The recording methods used in such studies have involved metal recording or ground/reference electrode(s) which can have their own complex responses when exposed to RF fields. The presence of metal in the preparation may produce unexpected effects, making the results difficult to interpret[13,14,26,27,36–39].

In the work presented here, we examined the instantaneous (and non-thermal) effects of RF exposure on the ongoing activity of neurons, using both electrical (with metal-containing silicon probes) and (metal-free) optical recoding of neural activity in awake rodents, providing a direct comparison of the two approaches. For RF-interference-free optical recording of neural activity, we developed a novel fiber-coupled head-mount 1-photon imaging set-up. While neural activity seemed to be affected by RF stimulation when monitored by electrodes, no neuronal responses were detected with optical monitoring to stimuli of even higher intensity.

## Results

We performed experiments to study the immediate effects of RF energy on neural activity in-vivo conducted via two distinct approaches: 1) by electrophysiological recordings, which involves metallic recording elements exposed to the RF energy, and 2) by metal-free optical 1-photon imaging.

Different RF antennas were used and driven by a simple RF circuit consisting of a software-tunable signal generator and a narrow band high power amplifier (Fig. 1a). The experiments were performed either inside a Transverse ElectroMagnetic (TEM) cell antenna (Fig. 1b) or using a patch antenna (Fig. 1c, d; see Methods). We selected 950 MHz as our operating frequency since this frequency can be delivered by affordable commercial equipment and is absorbed robustly while penetrating to a

sufficient depth in the rodent head. 950 MHz is also close to one of the Global System for Mobiles (GSM) frequency bands and is therefore of interest for investigations of the potential biological effects of mobile phone communication.

**Electrophysiological monitoring of transcranial RF-induced neuronal responses.** Rats and mice were implanted with multi-shank, multi-site metallic-electrode silicon probes[40] into either the pyramidal layer of the CA1 region of hippocampus or the parietal cortex. The micro-drive holding the probe, the connectors and head stage signal multiplexer/amplifier were shielded by a copper wire mesh on the head of the animal. Our temperature measurements showed that the RF power levels used in our electrophysiological experiments are thermally safe for the brain (Suppl. Figure 1). In addition, they confirmed that the head-cap, including the copper mesh, was not blocking the penetration of RF into the soft tissue, skull and brain from both sides of the head (Suppl. Figure 1j). Small daily movement of the micro-drive allowed us to survey different populations of neurons in subsequent sessions. Sessions consisted of ≥30 min of intermittent (2–3 s RF-ON followed by 2-3 s RF-OFF periods; Fig. 1e) RF radiation resulting in ≥100 of trials for RF-ON and RF-OFF comparisons. This intermittent stimulation strategy was chosen to control for the spontaneous brain state changes[40]. Neurons were clustered and separated into putative pyramidal cell and interneuron classes, using established physiological criteria[41]. Although transient RF-induced artifacts (at the RF onsets and offsets) were present in the local field potentials (LFP) (Suppl. Figure 2), they did not affect the waveform or amplitude of the extracellularly recorded action potentials ('spikes'; Fig. 2aiii and biii). Because the effects of induced electric fields, and presumably that of RF exposure, on neuronal excitability depend on the spatial relationship and morphology of the neuron relative to the direction of the field[42], stimulation experiments were performed while the animal was immobile/sleeping in the same position. RF radiation induced either excitation or suppression in a subset of the recorded population with a preference for the latter. Induced spike responses were typically present immediately after stimulation onset (i.e., within 20-40 ms; the first 20 ms bins were blanked to remove occasional RF onset artifacts (Suppl. Figure 2b)) and were maintained during the RF-ON periods (Fig. 2aii and bii) throughout the entire course of the experiment (Fig. 2f).

To quantify RF-induced changes in neuronal firing, a nonparametric statistical hypothesis test (Wilcoxon rank-sum test) was used. Neurons with significantly decreased or increased firing rates ($p < 0.05$) were regarded as affected by RF exposure. Across all experiments (from 3 behaving mice and 2 behaving rats), out of 405 isolated hippocampal CA1 neurons 118 cells (29.1%) were affected by RF exposure (Fig. 2d; since qualitative differences were not observed between mice and rats, the results are combined). Of the affected neurons 71 (60.1%) had decreased activity during RF-ON periods versus RF-OFF periods (i.e. their activity was suppressed) and 47 (39.8%) were excited. Also, 33.1% of recorded cells from mice and 22.3% of those recorded from rats were affected with statistical significance. After classifying the isolated neurons into pyramidal cells and interneurons, we found that 25.9% of pyramidal cells and 40.4% of interneurons were affected by RF exposure. We also examined whether sinusoidal amplitude-modulated RF signals can phase-entrain neuronal spiking (in 2 behaving mice and 2 behaving rats). At various tested frequencies (5 Hz, 10 Hz, and 20 Hz), we found that the spikes of neurons were also phase-modulated ($p < 0.05$; Rayleigh test; Fig. 2g). In addition, repetitive stimulation led to occasional recruitment of large population spikes (Suppl. Figure 3).

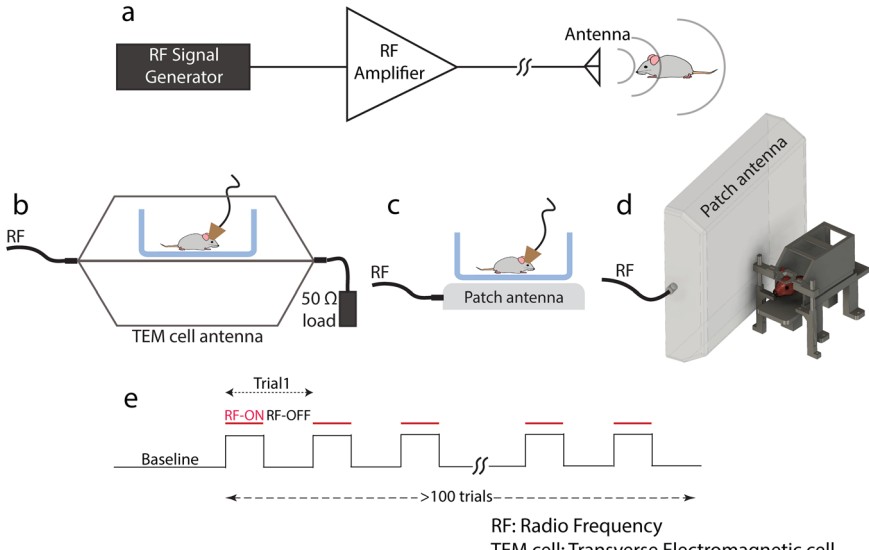

**Fig. 1 Experimental set-up for transcranial RF stimulation of the rodent brain. a** Circuit diagram for RF stimulation at 950 MHz. **b** Freely behaving mice/ rats in their home cages are placed between the parallel plates of the TEM cell antenna. **c** Freely behaving mice/rats in their home cages are placed on top of a patch antenna (b and c were used in electrophysiological recording experiments). **d** Head-fixed awake mice are irradiated by an adjacent patch antenna (with their head 4 cm from the surface of the antenna; used in 1-photon imaging experiments). **e** Timeline of an example session containing intermittent RF radiation (>100 trials).

Finally, we conducted dose-dependency experiments in the parietal cortical region in three urethane-anesthetized rats, which showed that the observed impact of RF depended on the magnitude of radiated power, and the same sets of neurons were activated at different power levels (27 affected cells out of 70 neurons; Fig. 2h and i).

A major concern with electrophysiological recordings is that the conductive metallic recording sites of the silicon probe implanted in the brain can act as a mini-antenna and amplify the electric field in nearby brain/tissue volumes[36,37,39]. As a result, neurons in the immediate vicinity of the iridium oxide ($IrO_2$) recording tips could be stimulated. Such an effect would clearly influence the measurements and would not exist in the absence of the implanted metallic components.

**1-photon endoscopic imaging of RF-induced neuronal responses.** In an attempt to eliminate the contribution of the 'metal-in-the-brain' effect, we used 1-photon optical $Ca^{2+}$ imaging (UCLA MiniScope (http://miniscope.org)[43], to record the neural activity of head-fixed mice. While $Ca^{2+}$ imaging does not directly measure neuronal spikes, it is a widely used methods for quantifying changes in "neuronal activity", which contains all aspects of activity involved in $Ca^{2+}$ changes, including spiking-related $Ca^{2+}$ influx. We applied intermittent RF stimulation (Fig. 1e) and compared the activity of neurons between the ON and OFF periods over ≥300 of trials. These experiments were performed in head-fixed mice to reduce variability due to the changing relationship between applied RF directions and neuronal orientation in the freely moving animals.

GCaMP6f was virally expressed in hippocampal CA1 neurons, followed by implantation of a 1.8-mm diameter graded-index (GRIN) optical lens[43]. Three weeks later, a 3D-printed plastic base (to hold the miniature microscope) and a head-fixing cap were implanted on the skull of the animal. Mice were gradually trained for the head-fixing procedure. During the imaging experiment, a patch antenna was placed 4 cm from the mouse's head and intermittent RF was applied. We limited the length of each session to 30 min to avoid photobleaching of the fluorescent signal. Because $Ca^{2+}$ transients are quite irregular with long tails and less frequent

than electrophysiological spikes, longer RF pulses were favorable. On the other hand, longer pulses limited the number of possible trials in each session. As a compromise, we chose 3s-ON 3s-OFF stimulation trials. Somatic $Ca^{2+}$ activity of isolated neurons was then compared between the ON and OFF periods over all trials.

Figure 3a illustrates the spatial distribution of the imaged CA1 pyramidal neurons in an example recording session in a head-fixed mouse. Figure 3b shows example $Ca^{2+}$ activity traces of sample neurons and Fig. 3c illustrates a transient change in the fluorescence signal associated with an isolated cell together with its computed deconvolved spiking activity (see Methods). The radiated power from the antenna, measured by a power meter (U2001A Power Sensor and N9912A Field Fox Spectrum analyzer, Agilent/Keysight) and a bidirectional coupler (778D, Agilent/Keysight), was set to 37.5 W. To quantify the responses to RF stimulation, we used the area-under-the-curve of the $Ca^{2+}$ transient traces and the deconvolved spiking activity derived from it as described earlier[44]. We used a bootstrapping analysis (comparing a given parameter driven from the original data versus the distribution of its value when driven from data with shuffled trial identities as previously shown[45] to compare the neuronal activity during RF-ON and RF-OFF epochs.

The results of our 1-photon experiment using the original UCLA Miniscope (V3) are summarized in Suppl. Figure 4, suggesting suppression of neural activity due to RF stimulation. The interpretation of these results were, however, questionable because when imaging with the original Miniscope device, we observed that RF exposure induced a decrease (1-4%) in the background florescence (Suppl. Fig. 5a). We identified the source of this artifactual effect as an interference caused by the RF radiation on the LED printed circuit board (PCB) of the Miniscope. Therefore, we modified the design of the device and replaced the LED PCB with a fiber-optic coupled to a high-power LED source (FCS-0470-200, Mightex; Suppl. Fig. 5c) placed inside a Faraday cage >5 m from the antenna (Suppl. Fig. 5f). In addition, we applied metallic shielding layers around the Miniscope's body and its transmitting cable (Suppl. Fig. 5g). After these modifications, background artifacts were completely eliminated (Suppl Fig. 5e).

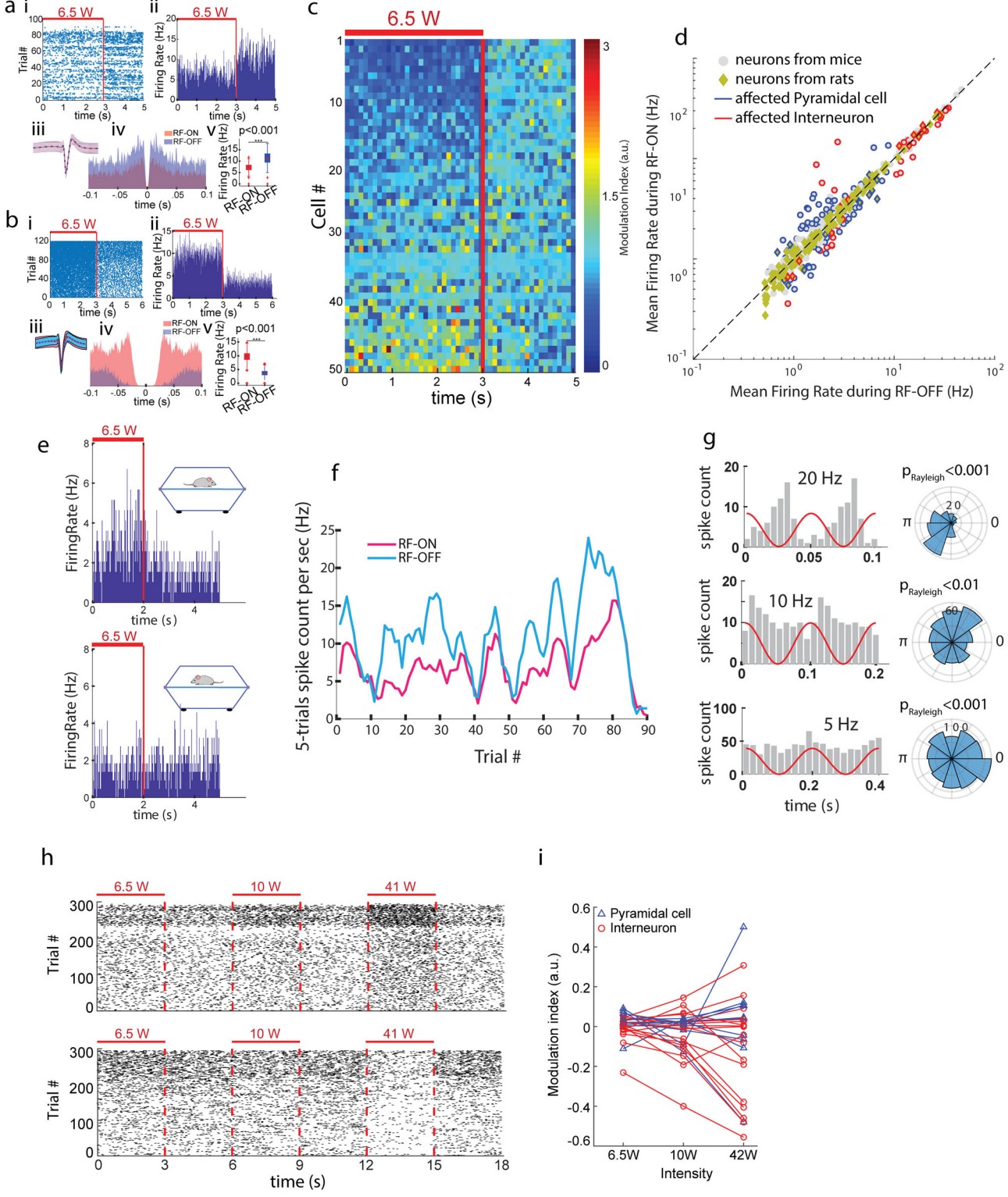

To address the concern that the added shielding near the animal's brain might affect the RF field patterns and potentially reduce the magnitude of the electric field, we measured RF-induced temperature increase in the brain using a similar shielding structure as in our stimulation experiments. We found that the RF-induced temperature effect (hence the absorbed RF electric field) was not attenuated but, in fact, was slightly increased by addition of the shielding material (Suppl. Fig. 6).

Applying this new system, our experiments with RF interference-free recordings showed no impact on Ca²⁺ activity of single neurons. Across all experiments (8 sessions in 5 mice), we found that 3.8% (51 of 1342) of CA1 neurons were significantly affected (either excited or suppressed) by the RF exposure (Fig. 3f). We repeated the same experiment in a Sham paradigm in which the animal was treated in exactly the same manner as in the Stimulation paradigm (Fig. 1d) but using an unconnected patch antenna. The same level of RF power as in the

**Fig. 2 Electrophysiological responses of hippocampal and cortical neurons in mice and rats to RF exposure. a** Example of spiking activity of an isolated single neuron in an immobile/sleeping mouse inside a TEM cell antenna (Fig. 1b) displaying suppressed activity by RF exposure. i: Raster plot of spikes for >80 trials of 3s-ON-2sOFF intermittent RF exposure. ii: Firing rate histogram (spikes/sec) of the same neuron. iii: Corresponding spike waveform (mean (line) +/- one standard deviation (shading)) during RF-ON versus RF-OFF epochs. iv: Spike timing autocorrelations for RF-ON versus RF-OFF epochs. v: Significant decrease of the mean firing rate during RF stimulation ($p < 0.05$; Wilcoxon Sign-rank test). **b** Same as a for another neuron with increased activity due to RF exposure. **c** Example of neural activity of simultaneously recorded neurons in recording session. Each row shows the color-coded firing rate of an isolated neuron normalized to the average firing rate of the same neuron in the RF-OFF epoch. Rows are sorted by the modulation index [(FiringRate_RF-ON-FiringRate_RF-OFF)/ (FiringRate_RF-ON+FiringRate_RF-OFF)]. **d** Significantly affected putative pyramidal cells (blue symbols) and interneurons (red symbols) are shown together with non-affected neurons (grey). Overall, out of the 405 neurons (in $n = 3$ mice and 2 rats), 118 (29.1%) were affected by the RF (activity of 71 (60.1%) cells were suppressed and that of 47 (39.8%) cells were excited). Also, 33.1% of recorded cells from mice and 22.3% of those recorded from rats were affected with statistical significance. Activity of 25.9% of the putative pyramidal cells (blue counters) and 40.4% of interneurons (red counters) were significantly affected by the RF exposure ($p < 0.05$; Wilcoxon sign-rank test). **e** The activity of same neuron is affected or not-affected by the RF depending on the position/direction of the animal with regard to the electric field. **f** Relationship between spontaneous firing rate fluctuation and RF response are kept throughout the recording period (same neuron as in a). Firing rates during RF-ON and RF-OFF periods along the whole session are shown separately. **g** Example of neurons whose spike firing are phase-entrained by amplitude-modulated RF at 20 Hz (top), 10 Hz (middle) and 5 Hz (bottom). **h, i** Dose-dependency of neural activity responses to RF energy exposure. **h** Dose-dependence of spiking activity (from an anesthetized rat) for a suppressed (top) and an excited (bottom) neuron. **i** Group statistics for dose-dependency of neural activity responses to RF energy exposure. Of the 70 neurons (in $n = 3$ anesthetized rats), 27(38.6%) were significantly suppressed or excited ($p < 0.05$; Wilcoxon Sign-rank test) by RF exposure. Note: due to the interference of the RF energy with the metallic electrodes in the measurement system, the interpretation of the results from the electrophysiological experiments is questionable.

Stimulation paradigm was applied to a second identical antenna placed >2 m away from the animal to mimic the same conditions (RF amplifier's noise) but without exposing the animal to RF energy. We found that under the sham condition, a similar small fraction of neurons (44 of 1293 = 3.4%) was affected (Fig. 3g). Comparison of neural responses for Stimulation versus Sham conditions revealed no significance difference ($p = 0.46$, Wilcoxon Rank-sum test) in the distribution of the modulation index, calculated using the sum of deconvolved spike heights (in 8 Stimulation and 8 Sham sessions in 5 mice; Fig. 3h). Similarly, no statistically significant difference was observed when comparing the percentage of cells (per session) that were significantly affected by the RF exposure (Fig. 3i). We examined additional statistical comparisons between Stimulation and Sham datasets using other variables: shuffle test on Spike Events (consisting of deconvolved spikes inferred from $Ca^{2+}$ transients but without considering their amplitude; Suppl. Fig. 7ai, 7bi), shuffle test on area-under-the-curve of the $Ca^{2+}$ transient traces (Suppl. Fig. 7aii, 7bii), and Wilcoxon rank-sum test on the amplitude of the deconvolved spikes (Suppl. Fig. 7aiii, 7biii). We did not observe any statistically significant difference in these tests.

In addition, in 3 mice, double sessions of RF and Sham stimulations were performed on two different days ≥48 h apart. In these animals, we combined data from the same imaged neurons across the two days and examined whether doubling the number of trials (≥600) would change the statistical results. Again, a very small subset of neurons (4.48% in Stimulation and 3.88% in Sham) was significantly affected (Suppl. Fig. 8a, 8b) and there was no difference between Sham and Stimulation conditions in the distribution of modulation index for all neurons (Suppl. Fig. 8c).

These results demonstrate that in the absence of implanted metallic electrodes, and at the power levels we could reliably investigate the effect of RF stimulation, neuronal $Ca^{2+}$ responses were not affected.

**Temperature measurements for estimation of energy deposition and electric field in vivo.** To monitor brain temperature changes, we used a metal-free optical temperature measurement system to avoid interference with the RF radiation. The tip of the fiber-optic temperature probe (1.1 mm diameter) was implanted in the brain in the same location that the GRIN lens for 1-photon imaging recording was placed (Fig. 4a). A 3D printed head-cap

for head-fixation was also mounted on the animal's skull. After recovery from surgery, each animal was trained for head-fixation for three days. During the experiment, the trained animals were head-fixed with the patch antenna placed in the same position as in the 1-photon $Ca^{2+}$ imaging experiments (Fig. 1d). We examined temperature changes in response to 3s-ON 3s-OFF RF exposure. With the exception of the highest power tested (68 W), we found that the temperature change in response to 3 s ON 3 s OFF pulses for 30 min remained in the physiological range as observed in a Sham session (35.5 °C to 39 °C; Fig. 4b, c;[46]). To quantify heat deposition, we applied continuous RF at different power levels for enough time to induce a reliable temperature increase (50 s) followed by 200 s off periods (Fig. 4d-f). We found a roughly linear relation between the radiated RF power and the induced temperature rise (in °C /s; Fig. 4g). The average temporal rate of temperature increase (ΔT/Δt) due to the RF exposure used in our imaging experiment at 37.5 W was 7.91e-3 °C/s ($n = 4$ mice).

Using the rate of temperature increase (Fig. 4g), we calculated the corresponding Specific Absorption Rate (SAR[47]; Fig. 4g) value in the brain of RF-exposed animals ($n = 5$ mice) and consequently deduced the magnitude of the local RF electric field (Fig. 4h). The SAR and E-field values for the applied RF in our imaging experiment at 37.5 W corresponded to 28.84 W/Kg and 231.77 V/m, respectively (ovals in Fig. 4g, h).

**Temperature measurements to evaluate the effect of metal-in-the-brain.** The optical temperature measurement also allowed us to directly evaluate the 'metal-in-the-brain' problem. Towards this goal, we implanted a thin (0.3 mm diameter) optical probe (PRB-100-01M-STM, Osensa Innovation Corp.) in the hippocampus, together with a triplet of tungsten recording electrodes (50 μm in diameter; Suppl. Fig. 9). Applying short (0.5 s) RF pulses of high intensity (68 W), we observed several tens of C° transient increase in brain temperature. After this first stage of the experiment, the metal electrodes were withdrawn from the brain and we applied the same RF energy exposure continuously for a longer period to ensure a clear reading of temperature rise (50 s) and measured the brain temperature in the absence of the metallic wires. Despite the very long RF energy deposition, brain temperature increased <2 C°, demonstrating the large impact of the metal electrodes on both the magnitude and the speed of temperature change.

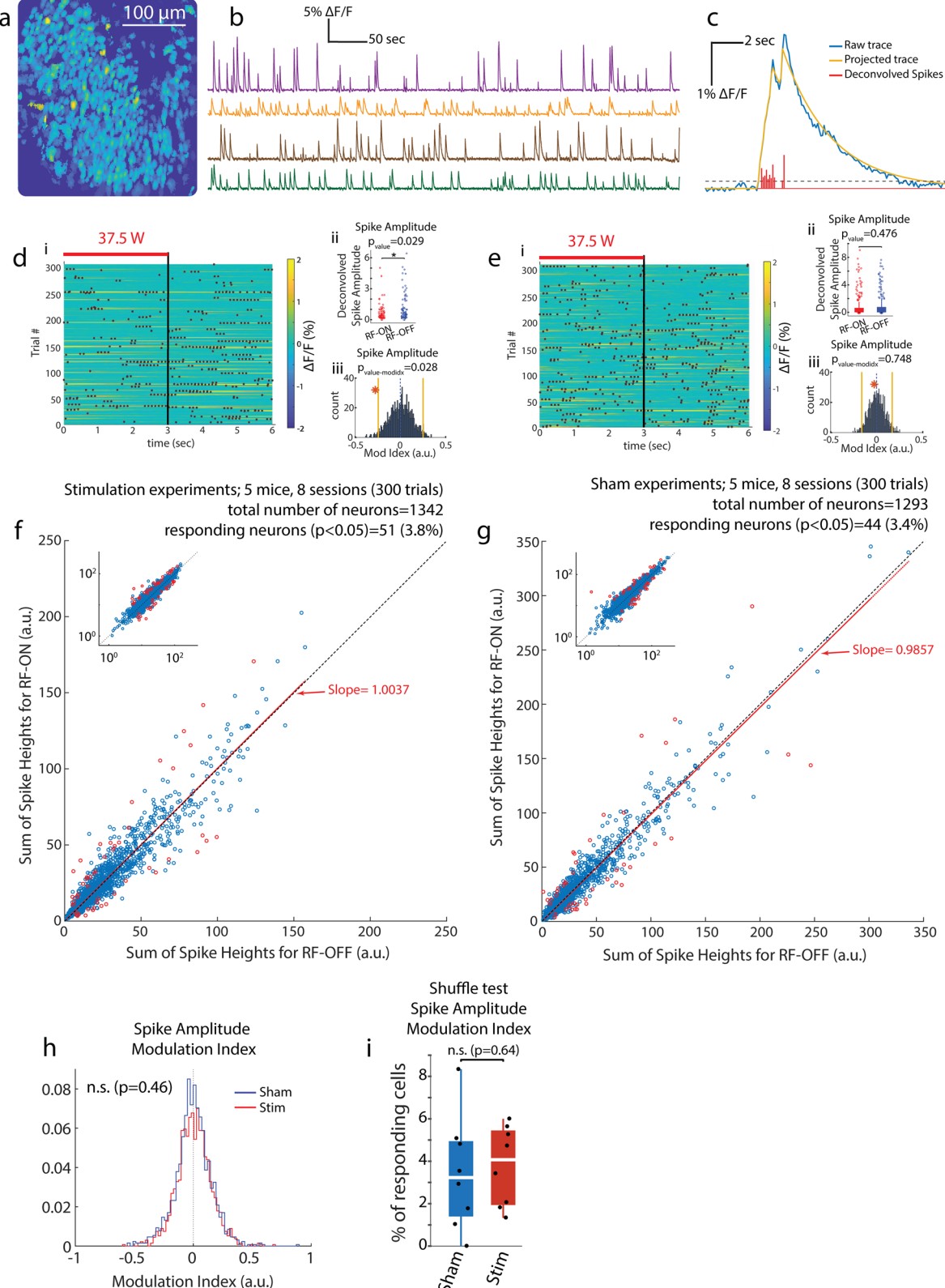

## Numerical simulations for assessment of electric field distribution in the brain

We conducted numerical simulations to assess the likely distribution of RF electric field inside the mouse head. Our simulations used a numerical mouse model derived from 3D Computed Tomography and cryosection[48], placed in a field-generating configuration and emulating the experimental set up (as shown in Fig. 1d). Suppl. Fig. 10 illustrates the distribution of the magnitude of the simulated electric field (Normalized to the maximum value) in three orthogonal sections of the mouse head at the same coordinates as the GRIN lens (for 1-photon imaging) and the temperature sensor implants. Note that the simulated E-field distribution in the region of the mouse brain (highlighted with a dashed oval) is relatively homogeneous, even though the field outside the brain can vary considerably in magnitude.

**Fig. 3 Metal-free, RF interference-free 1-photon Ca²⁺ imaging in mice. a** Field of view with spatial footprints of isolated single neurons from an example recording session. Brighter colors show higher activity neurons. **b** Example Ca²⁺ transient traces (ΔF/F where F is the intensity of the fluorescence signal) of four isolated neurons shown in different colors. **c** Illustration of an example Ca²⁺ original transient trace, its projected trace and putative deconvolved spikes (computed as reported earlier[44,74]). **d** Example neuron with statistically significant decreased activity during RF-ON versus RF-OFF epochs: i) heat-map of the Ca²⁺ transient trace with superimposed deconvolved spike events (black dots) during 300 trials. Red bar indicates RF stimulation. ii) Comparison of deconvolved spikes during RF-ON versus RF-OFF periods (Wilcoxon rank-sum test). iii) Shuffle test to compare the modulation index of the deconvolved spikes in the recorded session with 1000x random distribution of trial identities (computed as reported formerly[45]). **e** Same as in d but for a neuron with no statistically significant change in activity. **f** RF response of all recorded neurons (eight 30-min-long sessions with 3 s RF-ON 3 s RF-OFF trials in 5 head-fixed mice with patch antenna (see Fig. 1d), 37.5 W at 950 MHz). Activity of 51 out of 1342 neurons (3.8%) show statistically significant effect (shuffle test, $p < 0.05$) during RF stimulation (red circles). Inset shows the same plot in logarithmic scale. **g** The fraction of statistically significantly affected neurons was similar in sham stimulation sessions: in eight 30-minute-long sessions with 3 s RF-ON 3 s RF-OFF trials in 5 head-fixed mice with non-connected patch antenna (37.5 W at 950 MHz is fed in to a secondary antenna >2 m away). Activity of 44 out of 1293 neurons (3.4%) show statistically significant difference (shuffle test, $p < 0.05$; red circles). Inset shows the same plot in logarithmic scale. **h** Comparison of distribution of modulation index, calculated using sum of amplitude of deconvolved spikes inferred from Ca²⁺ activity, between Sham and RF stimulation in all experiments (1342 neurons from Stimulation and 1293 neurons from Sham; 30 minutes of 3 s RF-ON, 3 s RF-OFF trials; 8 Sham and 8 RF stimulation sessions in 5 mice). **i** Comparison of percentage of RF responding cells (per session) with statistically significant ($p < 0.05$) effect between Sham and RF stimulation in same data as in h.

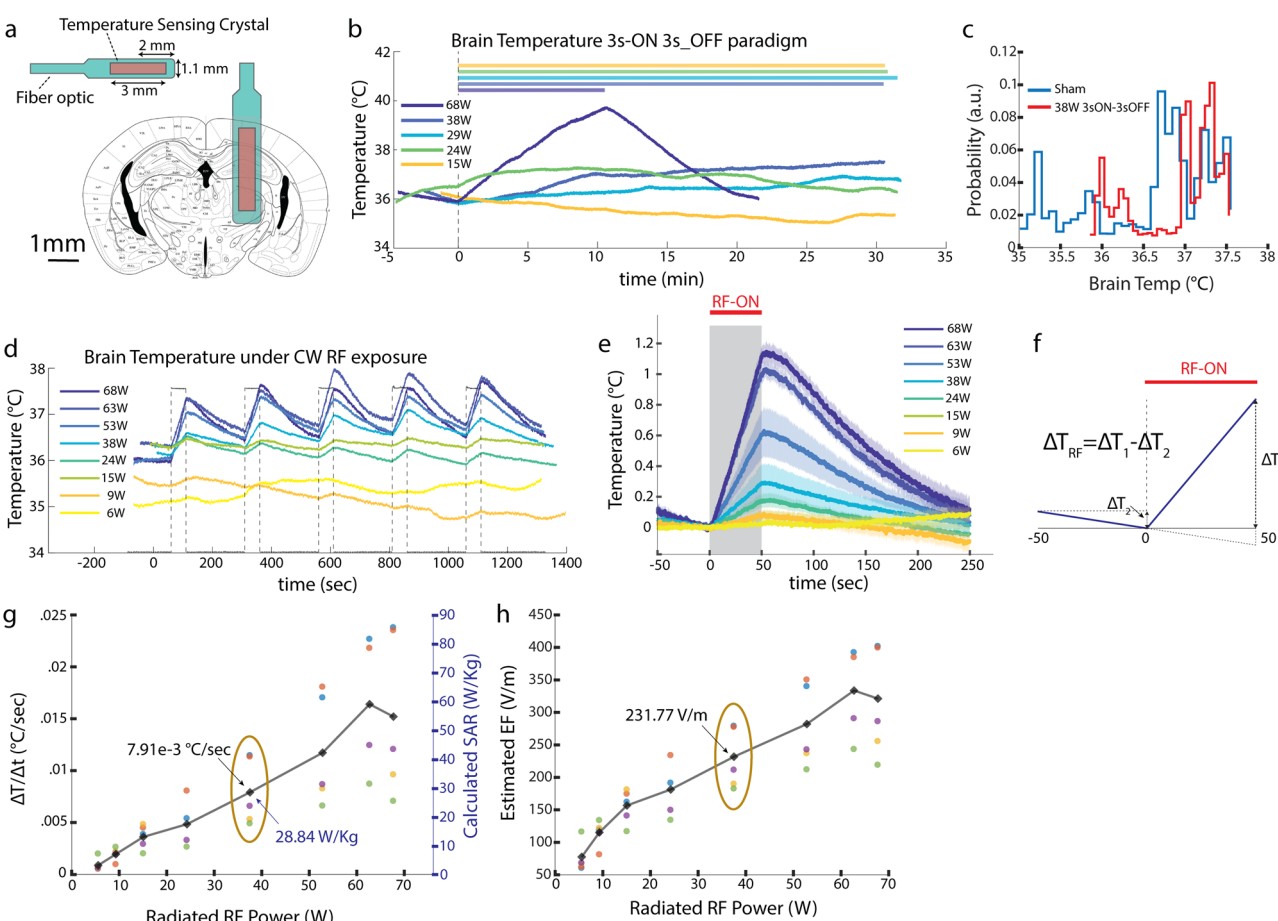

**Fig. 4 RF-induced brain temperature increase and estimation of specific absorption rate (SAR) and electric field (E-field). a** Illustration of the optical temperature sensor's dimensions and its implant location in mice brain to measure RF-induced temperature changes around hippocampal areas. **b** Brain temperature change in example 30-minute sessions with intermittent 3 s ON-3 s OFF RF stimulation, of various power levels, in a head-fixed (Fig. 1d) mouse (bar: RF ON). The highest power (68 W) session was terminated after 10 minutes due to rapid temperature rise. **c** Distributions of brain temperatures in RF (30 min of 37.5 W 3 s ON-3 s OFF) versus sham sessions (3 sessions of 30 min in the same mouse). Note that brain temperature stayed within the physiological range (deduced from Sham sessions) during RF stimulation. **d** Brain temperature changes in a head-fixed mouse in response to continuous-wave RF exposure with varying RF power (similar setting as in 1-photon experiments, Fig. 1d), **e** Brain temperature rise during 50-s RF stimulation (mean (line) and standard error of the mean (shade) from same data as in d). **f** Illustrative temperature recording describing how the net temperature change (ΔT) in response to RF is calculated taking into account the ongoing local temperature trend before stimulation onset (if the temperature changes in the 50 s prestimulus is not linear that stimulation trial is not taken into account for the calculations). **g** Left axis: Rate of brain temperature change in response to radiated RF (in 5 head-fixed mice (as per Fig. 1d); colors represent data from different animals; oval indicates the 37.5 W power level applied in the 1-photon imaging experiment in Fig. 3). Right axis: Calculated SAR value using temporal temperature change rate in e (see Method). **h** Estimated E-field values using SAR values in g (see Method). Different colors in g and h represent data points obtained from different animals.

## Discussion

We developed a fiber-coupled head-mount 1-photon imaging system and performed artifact-free optical neural recording in the brains of drug-free behaving rodents exposed to RF energy. Our experiments yielded relatively homogeneous in-situ electric fields of ~230 V/m, corresponding to SAR values of ~29 W/kg. Our results demonstrated that at this intensity, RF stimulation-induced brain temperature changes within the range of physiological variations but did not affect the ongoing neuronal activity as measured by 1-photon, single-neuron $Ca^{2+}$ imaging. We also report that, using lower levels of RF power, spiking activity was reliably affected in a fraction of hippocampal neurons when metal-containing silicon probes were used instead. We attribute the changes in spiking activity to indirect effects mediated by the local enhancement of RF fields by the presence of metal in the brain, which we showed directly with optical temperature measurements. Additionally, our experiments highlight the many pitfalls of measuring RF-induced effects in the brain of behaving animals, and offer guidance for future directions.

**Non-thermal effects of RF stimulation**. The "non-thermal" vs. "thermal" debate about the effects of electromagnetic fields on biological systems has a long history in microwave and radio-frequency radiation research[4–9]. Using a modified fiber-coupled version of the UCLA Miniscope, our interference- and artifact-free experiments provide a ground truth for this question. Our results show that RF radiation does not modify ongoing neural activity of hippocampal CA1 neurons, as measured by $Ca^{2+}$ imaging, in an immediate and non-thermally mediated manner for in-situ electric fields up to ~230 V/m. Our optical temperature measurements showed that RF-induced temperature changes remained within physiological limits, although the exposure was above the levels recommended by regulatory agencies (e.g. US regulations limits, set by the Federal Communication Commission[49], are 1.6 W/Kg and 8 W/Kg, averaged over 1 gr of tissue, for public and occupational/controlled cell phone exposures, respectively). With our system, we could not test neuronal responses beyond 40 W RF power level due to limits in the tolerance of electronic circuits in our imaging instrument to high RF power. However, future modification of the Miniscope (or development of high-quality fiber-bundle-based imaging systems) may alleviate this technical problem. Although our results do not exclude non-thermal RF-induced effects on neural activity, they do rule out such a possibility for exposures levels higher than regulatory limits in real-life scenarios.

**Technical challenges for determining RF-induced physiological changes**. We examined the possibility of non-thermal effects of RF energy on neural activity using both electrophysiological recordings and 1-photon $Ca^{2+}$ imaging. It has been shown repeatedly that changes in $Ca^{2+}$ transients do not follow an exact match with the neuron's sodium spiking activity[50]. Even in in-vitro preparation, single action potentials are rarely resolved. Instead, $Ca^{2+}$ transients correlate much better with spike bursts, which are known to involve $Ca^{2+}$ influx[51]. Yet, $Ca^{2+}$ imaging (whether 1-photon or 2-photon) methods are still considered a reliable measure of neuronal activity and constitute the state-of-the-art optical recording methodology. The term "neuronal activity" is typically vaguely defined and can vary from spikes in pyramidal cells and inhibitory interneurons, excitatory post-synaptic currents (EPSCs), inhibitory postsynaptic currents (IPSCs), $Ca^{2+}$ spikes, plateau potentials, etc. Thus, $Ca^{2+}$ imaging may be considered as a method more sensitive to measure "neuronal activity" of various kinds, not only spiking.

Considering these arguments, we could not examine a quantitative and head-to-head comparison of the results from the two methods.

Our electrophysiological experiments highlighted an instantaneous and stable change in neural activity in a considerable fraction of cells in both hippocampal and parietal cortical regions of mouse and rat brain. The fractions of the affected neurons were comparable to those reported during transcranial electrical stimulation[40,52]. Pyramidal cells and interneurons were similarly affected, and both excitatory and inhibitory responses were observed, with a preference for suppression of activity. RF effects on spiking activity of neurons, monitored by a multi-electrode array, have been observed previously in tissue culture exposed to both continuous-wave and Global System for Mobile (GSM) signal at 1,800 MHz (0.01 to 9.2 W/kg SAR). The suppression of burst spiking was present for both exposures and was greater with GSM signal[39,53]. In hippocampal slices in vitro, population spikes were enhanced during exposure to 700 MHz continuous wave RF fields[36], similar to our electrophysiological observations (Suppl. Figure 3). Furthermore, it has been reported in both humans and experimental animals that exposure to weak electromagnetic fields at levels emitted by cell phones can influence brain electrical activity[24,37,54–57]. Common in all these experiments is the presence of a metal conductor in connection with brain tissue (EEG electrodes on the scalp, in human studies). Even when glass pipettes are used for recording, the metallic ground or reference electrode is exposed to the applied RF fields. Our experiments offered a head-to-head comparison between the presence and the absence of metal in the brain. We found that using the same frequency and lower power levels of RF exposure, recording with metal-containing silicon probes reliably affected firing patterns of neurons, whereas using optical recordings demonstrated absence of any reliable effects. We attribute the neural effects observed in electrophysiological experiments to the interference of RF fields with the exposed metallic electrodes.

The mechanisms by which the presence of a metal conductor, exposed to RF energy, can cause such changes in neural signals is not fully understood[26,36–39]. It is well known that electric fields can become concentrated outside the surface of conductors, particularly in regions of high curvature (like the points of lightning rods or sharp electrodes). Our control experiment measuring the thermal effect of presence of metal in the brain suggests that local enhancement of the electric field in the vicinity of metal electrodes enhances heat production in that region, which can in turn affect ongoing neural activity[58,59]. In our electrophysiological experiments, neurons reliably followed at least up to 20 Hz sinusoidal stimulation. Thus, one might argue that this finding favors a nonthermal mechanism because changes of local temperature could not have occurred fast enough to be responsible for the spike entrainment. On the other hand, our direct measurements of brain temperature with and without electrodes in the brain showed that the presence of metal can dramatically increase both the magnitude and speed of temperature change (Supp. Fig. 9). Such local temperature increase can reach several tens of C°, which can effectively alter firing patterns of neurons without necessarily damaging them, provided that the product of the magnitude and duration of RF stimulation remains below a certain threshold. In our electrophysiological experiments, we were able to drive or silence neuronal spiking activity repeatedly for many hours and across multiple sessions without apparent long-term effects on neuronal activity. Thus, it is possible to find a 'safe but effective' range of RF deposition for non-invasively affecting neuronal activity in this experimental set-up.

An important conjecture from our finding is that while low power RF exposure levels that are allowed by regulatory limits

may not affect neurons in the intact brain, patients with metal implants in the brain and body may be vulnerable. The large amplification effect we observed is consistent with concerns about risks to patients with metallic implants might face during MRI examinations[60] or other occasions of exposure to high intensity RF energy.

**Further directions**. While our findings address unsolved challenges in assessing direct effects of continuous-wave RF exposure in vivo, they do not exclude potential responses to pulsed RF stimulation using very high power short-time RF pulses. Short (10-100 μs) high energy RF pulses induce sound perception in both humans and experimental animals[61,62]. Because the estimated temperature changes associated with such short pulses are only on the order of $10^{-6}$ °C[62], it has been suggested that vibration-mediated thermos-elastic effects in brain tissue are the principal mediating effect[62–64]. In addition, subsequent experiments in cats (918 or 2450 MHz) showed that destruction of the cochlea eliminated the brain evoked potentials, suggesting that the audible clicking sensations are due to mechanical vibration[65,66], similar to indirect ultrasound pulse-induced effects[67,68]. This potential mechanism related to pulsed RF (e.g., GHz RF signal modulated by kHz pulses) justifies the value of future studies replicating our experiments for pulsed RF exposure.

In addition to its potential for deleterious effects, RF radiation may also have unexpected upsides. If its neural effects were to be documented clearly and shown to be controllable, RF radiation could constitute a new tool for non-invasive transcranial stimulation of the intact brain, whether for scientific investigation or for therapeutic purposes. It is well-demonstrated that thermal changes do exert consistent effects on neuronal firing patterns even within the physiological temperature range[46,58]. Identifying a safe zone of power exposure and stimulation patterns could make RF energy a useful tool, similar to transcranial ultrasound[9,67]. Thus, while our results demonstrate that the weak RF fields induced by cell phones are unlikely to exert an immediate effect on neural activity, our failure to modify neuronal circuits in the experiments reported here should not be taken as a wholesale rejection of the concept of RF stimulation. Harnessing RF energy, while controlling RF-induced thermal changes to remain within safe limits, may still prove to be a useful tool for non-invasive brain stimulation.

Finally, we emphasize that our results do not rule out all potential harmful effects of RF radiation on neurons or more generally, the central nervous system. Even at the exposure levels used in our experiments, it is possible that applied RF fields might affect the neurons chronically, or in other aspects not explicitly considered here[3,5,34].

## Methods

**Animals**. Adult wild-type C57BL/6JxFVB mice (20–35 gr) were obtained from Charles River Laboratory. Mice were kept in cages in a 12 h regular cycle vivarium room dedicated to mice in up to five-occupancy cages.

Adult Long-Evans rats (250–450 gr) were obtained from Charles River Laboratory and kept in a 12 h regular cycle vivarium room dedicated to rats in double-occupancy cages.

In all experiments, each animal served as its own control, no randomization or blinding was employed. No prior experimentation had been performed on the animals.

Implanted animals (mice and rats) were moved to single occupancy cages to avoid both social conflicts between animals and damage to the implanted materials.

**Mice head-fixation**. A mice head-fixation set-up with a matching head-mount cap (Fig. 1d) was designed and 3D printed using a resin-based printer (Form2, Formlabs Inc.). Design files are available here: (https://github.com/omidyaghmazadeh/3D_Print_Designs/tree/main/Mouse_HeadFixation_SetUp). This set-up is used for 1-photon calcium imaging, and related temperature measurements. Before starting

the experiments, animals were trained to habituate with head-fixation for incremental duration from 5 min to up to 2 hrs.

**Transcranial radio frequency stimulation**. Radiofrequency wave at 950 MHz were radiated from commercially available antennas: either a patch antenna (M3070100P11206-B, Vente/TerraWave) or a TEM cell antenna (TBTC3, Tekbox Digital Solutions). The RF signal was generated by a compact USB-controlled signal generator (SynthHD, Windfreak Tech. LLC) and amplified by a narrowband high power amplifier (ZHL-100W-13 + , Mini-circuits), then fed into the desired antenna. In the case of RF stimulation with a Patch antenna, the antenna is placed either next to animal's home cage (Fig. 1c) or next to the head-fixation set-up in a fixed position relative to animal's head (Fig. 1d). In the case of TEM-cell antenna animal's metal-free plastic homepage is placed between the walls of the antenna far from its edges (Fig. 1b).

**Silicon probe implantation for Electrophysiology in freely behaving mice and rats**. Silicon probe implantation were performed as described in former reports[40,69]. Mice or rats were kept anesthetized under a steady stream of isoflurane (2%) and craniotomies were performed under stereotaxic control. The rectal temperature was kept constant at 36–37 °C with a DC temperature controller (TCAT-2, Physitemp LLC) and stages of anesthesia were maintained by confirming the lack of vibrissae movements and nociceptive reflex. Mice and rats were implanted with 32- or 64-site silicon probes (NeuroNexus or Cambridge Neuro-Tech) in the pyramidal layer of the CA1 region of the hippocampus (AP −2 mm, ML + 1 mm, DV -1 mm for mice and AP -3.5 mm, ML 2.5 mm, DV -2.25 mm for rats). A pair of reference (stainless steel) wires were bilaterally implanted through the skull above the cerebellum, and a grounded copper mesh hat was constructed, shielding the probes. The reference wires/screws were then connected to the ground wires of the probe. Probes were mounted on in-house made 3D printed micro-drives that were gradually advanced to reach the targeted CA1 pyramidal layer over the course of 2–5 d after surgery. Hippocampal neural layers were identified physiologically by characteristic LFP patterns[41]. Animals were allowed to recover from surgery before conducting any experiments.

**Electrophysiological recording in freely behaving mice and rats**. 3 adult mice and 2 adult rats were used in these experiments. Mice inside their home cages and rats inside a dedicated experiment cage, are placed in the space between the parallel plates of the TEM cell antenna. Any metallic parts are detached from the cages prior to the experiment. Electrophysiological recordings were performed using an Intan RHD2000 interface board (Intan Tech.) and 64-channel digital head-stages (Intan Tech.) at a sampling rate of 20KHz. Voltage from recording electrodes were measured in a unipolar manner against the ground/reference electrode (implanted in the cerebellum). The TEM cell antenna is fed with 950 MHz amplified RF signal once the animal is asleep or immobile in a sleep posture. RF stimulation is applied in 5 s trials (consisting of 3sON-2sOff or 2sON-3sOFF). Animal's posture is monitored to make sure animals head position with regard to the direction of the RF field is immobile. If the animal changes the position of its body and/or its head before the period of immobility reaching ~100 trials, data is considered not useful, and the recording would start over in a new sleep posture. In a special set of recording sessions, amplitude-modulated RF wave is applied (950Mhz, amplitude modulated at 5, 10, or 20 Hz, 2sON-3sOFF).

**Silicon probe implantation for electrophysiology in urethane-anesthetized rats**. Adult, male Long-Evans rats ($n = 3$) were used for the anesthetized, extracellular electrophysiology experiments. After anesthesia induction (isoflurane, 2%), urethane (1.3–1.5 g/kg, intraperitoneal injection) and atropine (0.05 mg/kg, subcutaneous injection) were administered. The rectal temperature was kept constant at 36–37 °C with a DC temperature controller (TCAT-2, Physitemp LLC). Stages of anesthesia were maintained by confirming the lack of vibrissae movements and nociceptive reflex. Skin of the head was shaved, and the remaining fur was completely removed by using depilatory cream. The skin was retracted after a mediosagittal incision, and the bone surface was cleaned with hydrogen peroxide (2%) and dried. Metabond (Parkell Inc) was applied on the skull surface and rats were implanted with a custom-designed, 3D-printed plastic baseplate[69]. Craniotomy was drilled (1 mm diameter) and a silicon probe (Buzsaki 5×12; NeuroNexus) attached to a micro-drive[69] was implanted at 2 mm posterior from bregma and 3.8 mm lateral of the midline at 18 degrees, in the cortex. The hole around the probes was filled with nonconductive silicon (Dow Corning). A stainless-steel screw was placed above the cerebellum and served as ground.

**Electrophysiological recording in urethane-anesthetized rats**. After the implantation of the probe in the acute surgery, the animal was placed inside a TEM cell and the depth of the silicon probe was fine-tuned using the micro-drive. Once the target depth (4 mm from surface of the brain) was reached, we waited at least 30 minutes before data collection. We recorded 30 min pre-stimulation baseline, 300 trials of intermittent RF stimulation (three different intensities were applied at 950 MHz, 5.4, 8.9 and 37.5 W, respectively; 3 s RF-ON was followed by 3 s RF-OFF epoch for each intensity) and 30 min post-stimulation baseline. The recorded signals ($n = 64$ channels) were amplified and stored after digitization at 20 kHz

sampling rate per channel (RHD USB Interface Board, Intan Technologies). After the experiment, the silicon probe was recovered and cleaned in distilled water. Animals were humanely euthanized after the end of the post-surgery experiments.

**Fiber-coupled UCLA miniscope for RF-artifact-free 1-photon Ca²⁺ recording.** We observed that the LED source of the UCLA Miniscope-V3[70] is affected by the RF wave of our stimulation paradigm leading to a reduction of the background fluorescence in the Miniscope images in the presence of RF waves (Suppl. Fig. 5a). In order to remove this artifact from our measurements we removed the LED-PCB of the Miniscope system and provided the required light by a 3 mm wide optic fiber (FPC 960/1000/2200-0.63_8.0 m FCM-COL(3 mm), Doric Lenses Inc.) connected to a high power LED source (FCS-0470-200, Mightex) put inside a faraday cage (made of 4 mm thick aluminum sheets) placed 5 m away from the antenna (Suppl. Fig. 5f). Images obtained by the fiber-coupled scope are artifact free (Suppl. Fig. 5e) but the system itself occasionally stopped working when exposed to higher levels of RF fields. To reduce susceptibility of the Miniscope CMOS camera circuitry to RF, the modified (fiber-coupled) Miniscope is then covered by a layer of copper adhesive tape and a layer of aluminum foil for shielding from the RF radiation. The shielded scope is then placed on top of the 3D printed based attached to animal's skull. The Miniscope's coaxial cable is replaced by a longer (6 m) cable to keep the Miniscope data acquisition system (DAQ) also far away from the RF source. The coaxial cable is also shielded by stainless steel metal braided sleeving. These modifications permitted us to record non-interrupted artifact-free 1-photon Ca²⁺ imaging with relatively high level of RF exposure (up to 40 W input power into the patch antenna in our recording set-up (Fig. 1d)). The recordings are performed at 30 frames per second rate and saved in fixed-time (50 s) length ".avi" format files. The UCLA Miniscope software also permits simultaneous recording of an additional video mainly for the propose of animal behavioral tracking. These videos are recorded with the same frame rate and saved in a similar format as the actual imaging. In our experiments we used this channel to record a simultaneous video (using a commercially available webcam (LifeCam Studio, Microsoft)) of an LED indicating the stimulation periods to synchronize the imaging data with the stimulation timing.

**Surgery for Ca²⁺ imaging: virus injection, GRIN lens implantation and base mounting.** The surgical preparation for Miniscope recording were made in two steps, separated by three weeks, following previously published methods[43]. In the first surgery, mice were anesthetized with isoflurane and craniotomies were performed under stereotaxic control. Following a small craniotomy, 500ul of GCaMP6f virus (pAAV.Syn.GCaMP6f.WPRE.SV40 (AAV1), obtained from Addgene) was injected at the pyramidal layer of CA1 region of hippocampus (AP -2.1 mm, ML + 2 mm) and at 1.65 mm depth from the surface of the skull. Twenty minutes past the end of the injection a second bigger (~1.8 mm in diameter) craniotomy was made to accommodate the implantation of a GRIN lens (1.8 mm Diameter, 670 nm DWL, 0.0 mm WD, Uncoated GRIN lens, Edmund Optics). Cortical tissue was aspirated by a custom-made vacuum system and neural fibers in the corpus callosum layers are carefully removed as described[43]. The GRIN lens, held by a 3D printed custom-made apparatus, was then aligned at the center of the craniotomy and gradually lowered to 1.45 mm depth from the surface of the skull. The lens was then secured to the skull by dental cement. The exposed side of the lens was then covered by a silicone adhesive paste (KWIK-SIL, World Precision Instruments). In the second surgery, three weeks after the first one, mice were again anesthetized with isoflurane. The silicone adhesive was removed and UCLA Miniscope-V3 with a 3D printed base attached to it was carefully held on top of the GRIN lens and moved to find the best position to obtain a field of view with the highest number of cell counts. Once the optimal position was obtained the base was fixed to the skull with dental cement[43].

**1-photon calcium imaging miniscope recording.** Mice were head-fixed at a constant location in reference to the patch antenna (Fig. 1d). 1-photon calcium imaging was recorded using the modified fiber-coupled UCLA Miniscope as described above. Each imaging sessions was limited to 30 min of duration to ensure imaging quality. RF radiation, composed of several trials of intermittent RF-ON and RF-OFF, was applied. While shorter trial length (i.e. larger number of trials) were beneficial to gain statistical power, longer trials would help inclusion of higher number of events per trial. We found that 3 s ON-3s OFF trials would provide a good compromise. Therefore, for stimulation sessions, 3 s ON-3s OFF RF energy at 37.5 W at the antenna input was applied and 30 minutes long sessions were recorded. Sham sessions were recorded with exactly similar set-up and parameters as in the stimulation sessions with the difference that the RF was fed into an antenna which was placed more than 2 m away. This ensured inclusion of the sound noise generated by the RF amplifier in the sham paradigm. Two stimulation sessions and two sham sessions, with 48 h time-space (to ensure better imaging quality), were recoded from the same animal for double-sessions data. Animals were humanely euthanized after the end of the experiments.

**Temperature probe implantation.** Mice and rats were anesthetized with isoflurane and the implantation was guided by stereotaxic control. After skin incision and skull cleaning a custom-made 3D printed plastic base (Clear04 resin,

Formlabs) was glued to the skull using dental cement (C&B Metabond, Parkell Inc.). Metal-free optical temperature probes (OTP-M, OpSens Solutions Inc., or PRB-100-01M-STM, Osensa Innovations Corp.) were implanted in mice at (AP −2.1 mm, ML 2 mm, DV 3.75 mm) for temperature measurements related to 1-photon imaging experiments ($n = 5$ adult C57BL/6 mice), at (AP −2.1 mm, ML 1.5 mm, DV 1.25 mm) for metal-in-the-brain effect experiment ($n = 1$ adult C57BL/6 mice) and for shielding effect experiment ($n = 1$ adult C57BL/6 mice), and at (AP -2 mm, ML 2 mm, DV 4 mm) for those related to electrophysiological experiments ($n = 4$ adult C57BL/6 mice) through a 1.8 mm craniotomy (Fig. 4a and Suppl. Figure 1c). For rats ($n = 4$ adult Long-Evans rats) the temperature probe was implanted at (AP 2 mm, ML 2 mm, DV 4 mm; Suppl. Figure 1b). The hole around the sensor was filled with a nonconductive silicone elastomer (Kwik-Sil, WPI Inc.). A custom-made 3D printed cap was attached to the base in order to protect the sensor. Animals were allowed to recover from surgery before conducting any experiments.

**Brain temperature recording related to electrophysiological recording and 1-photon imaging.** For these experiments a 1.1 mm diameter optical temperature probe (OTP-M,Opsens Solutions Inc.) was used. Temperature recordings were collected from the analog outputs of the specific signal conditioner (AccuSens, Opsens Solutions Inc.) and recoded by a data acquisition board (RHD USB Interface Board, Intan Technologies) to ensure synchronous recording with the RF stimulation trigger signal.

For temperature measurements related to the 1-photon imaging experiments (Fig. 4), the recording was made with head-fixed animals with exact configuration as for the RF stimulation as shown in Fig. 1d. Brain temperature is recorded for baseline, 50 s stimulation period with continuous RF radiation at various power levels followed by post-stimulation RF-Off periods.

For temperature measurements related to the electrophysiology experiments (Suppl. Figure 1), animals were put in a mouse cage to fit in the TEM cell antenna. Recording was started immediately after the animal fell asleep. Different duty cycles (10, 20, 30, 40 and 50 percent, $n = 60$ trials each) of 5 s period were applied at different intensities. 5 min baseline were recorded before and after each stimulation period. To mimic our electrophysiological recordings, we also applied 300 trials of 40% duty cycle of 5 s period at different intensities. 15 min baseline were recording before and after stimulation. To improve signal-to-noise ratio in our recordings, we also applied 150 s, continuous wave TRFS at different intensities, interleaving with 10 min baselines.

**Brain temperature recording related to shielding and metal-in-the-brain effects.** For these experiments either a 0.3 mm diameter optical temperature probe (Osensa Innovations Corp.) or a 1.1 mm diameter optical temperature probe (Opsens Solutions Inc.) was used. Temperature recordings were collected from the corresponding specific signal conditioner (FTX-300-LUX +, Osensa Innovations Corp. or AccuSens, Opsens Solutions Inc., respectively).

To examine if the presence of shielding of recording devices prevents absorption of the RF electric field by the brain we conducted two experiments addressing both electrophysiological recording and 1-photon imaging configurations.

To test whether the presence of a copper mesh shielding in the electrophysiological recordings affects the RF energy absorption in brain we implanted an optical probe (300 μm diameter, Osensa Innovations Corp.) into the hippocampus of a urethane anesthetized adult mouse. We applied three 50 s continuous wave pulses of RF energy (68 W) in an adjacent patch antenna. Then, we placed a copper-mesh head cap on animal's head and repeated a similar RF radiation pattern. Brain temperature increase due to RF exposure was measured both in the presence and absence of the wire mesh shielding (Suppl. Figure 1j). To test the effect of shielding applied in our 1-photon imaging we conducted a similar test but in a head-fixed awake mouse (implanted with a 1.1 mm diameter optical probe, OTP-M, Opsens Solutions Inc.). We created a similar shielding as in our 1-photon imaging experiments (consisting of a layer of copper tape covered by a layer of Aluminum foil) around the temperature probe and on top of animal's head. RF radiation is produced by a patch antenna placed next to the animal as in the configuration of Fig. 1d. Brain temperature increase to RF exposure was measured both in the presence and absence of the shielding (Suppl. Fig. 6).

For testing the impact of the presence of metal in the brains exposed to RF energy, we implanted an optical probe (300 μm diameter, Osensa Innovations Corp.) into the hippocampus of an urethane anesthetized adult mouse, along with a triplet of tungsten wires (50 μm in diameter, placed approximately 0.5 mm from the optical probe). The wire bundle is attached to the moving shuttle of a microdrive and the temperature probe was fixed to its wall. Animal was exposed to RF radiation from a nearby patch antenna (as in Fig. 1d). We applied the highest power (68 W) of RF stimulation in order to have a reliable and clear temperature increase for the metal implant-free situation. After few short pulses (0.5 s) were applied, the electrodes were withdrawn from the brain (with the temperature probe staying in its location) and the same procedure was repeated. Because in the absence of the electrodes in the brain the RF-induced power to a short pulse was very small, we applied a continuous RF exposure for 50 s (Suppl. Fig. 9).

**Spike sorting and unit classification for electrophysiological recording**. Single unit spike sorting was performed semi-automatically by Kilosort[71], followed by a manual curation using Phy software (https://github.com/kwikteam/phy) and custom-designed plugins (https://github.com/petersenpeter/phy-plugins) to obtain isolated single units. Cluster quality was assessed by manual inspection of waveforms and auto-correlograms. Putative excitatory and inhibitory neurons were separated on the basis of their waveform characteristics and the shape of their auto-correlograms[41].

**Extraction and cleaning of calcium traces for 1-photon Ca$^{2+}$ imaging**. Calcium activity traces of neurons were extracted from the recorded videos using the MiniscopeAnalysis package[72] (https://github.com/ettierguillaume/MiniscopeAnalysis) which itself is founded based on NormCorre (for image registration and alignment[73]) and CNMFE (for cell segmentation and activity extraction[44,74]) packages. For double-sessions analysis, tracking neurons across different days is done using the CellReg package[75].

The timing of the extracted Ca$^{2+}$ activity traces is corrected using the time stamps of the recorded 1-photon imaging frames (saved in a text file by the recording system). The extracted Ca$^{2+}$ activity traces are detrended and ΔF/F ($= \frac{F(t)-F_M}{F_M}$, where $F(t)$ is the fluorescence at time $t$ and $F_M$ is the median value of fluorescence for the entire session) traces are consequently calculated. In addition to the magnitude of the ΔF/F traces (area-under-the-curve), deconvolved spiking activity is also extracted, using the CNMFE package[44,74], and used in the analysis. Using a custom-made MATLAB (MathWorks) user interface the following manual curing steps are performed: 1) noisy traces and isolated neurons with abnormal body shapes are excluded, 2) a noise threshold is set to identify a lower limit for the detected spiking events from the Ca$^{2+}$ activity traces and 3) periods of noisy activity are registered for each Ca$^{2+}$ trace and are consequently eliminated from the data.

**Modulation index**. In both 1-photon calcium imaging and electrophysiological recordings the effect of the RF stimulation on the ongoing neural activity is measured by comparing the activity in the ON versus OFF periods over several number of trials. This comparison is made by calculating the modulation index (MI) on the measured neural activity of interest, such as rate of deconvolved spikes from calcium transients in 1-photon imaging or firing rate of spikes in electrophysiological recordings, etc., which is defined as follows:

$$MI = \frac{\left(measured\ parameter_{ON\ period} - measured\ parameter_{OFF\ period}\right)}{\left(measured\ parameter_{ON\ period} + measured\ parameter_{OFF\ period}\right)}. \quad (1)$$

**Specific absorption rate (SAR) and electric field (E-Field)**. SAR was calculated from temperature measurements with 50 s continuous-wave RF exposure (Fig. 4) using:

$$\rho C \frac{dT}{dt} = \nabla.(k\nabla T) + SAR\rho \quad (2)$$

where ρ is the tissue density (kg/m3), C is heat capacity (J/kg/C), k is thermal conductivity (W/m/C), and SAR (W/kg) is the driving force for temperature rise defined as: $SAR = \sigma|E|^2/2\rho$ (Eq. 3). Where E is the induced electric field (V/m), and σ is the electrical conductivity (S/m). When the heating duration is short, Eq. 2 is simplified to: $SAR = C\Delta T/\Delta t$ (Eq. 4) where ΔT/Δt is the temporal rate of temperature increase. SAR is calculated from the temporal temperature increase rate using Eq. 4 and EF is driven from SAR values using Eq. 3[47].

**Statistics and Reproducibility**. Statistical analyses were performed by MATLAB built-in functions or custom-made scripts. No statistical methods were used to predetermine sample sizes, but our sample sizes are similar to those generally employed in the field. All data presented were obtained from experimental replicates with at least three independent experimental repeats for each assay. All attempts of replication were successful. The unit of analysis were at single cell level. For many comparisons random shuffling of data (over trials) was applied and $p$ value was determined by distance from the distribution mean as reported formerly[45]. In other occasions, unless otherwise noted, nonparametric two-tailed Wilcoxon rank-sum (equivalent to Mann-Whitney U test) or Wilcoxon signed-rank test was used. Due to experimental design constraints, the experimenter was not blind to the manipulation performed during the experiment.

**Numerical simulations to assess SAR and electric field distribution in mice brain**. We used a commercial 3D electromagnetic simulation software (xFDTD, Remcom[76]) to perform numerical simulations to estimate the distribution of the fields in the mouse brain. The geometry of the patch antenna used in our experiments (M3070100P11206-B, Vente/TerraWave) was replicated in the software, and it was driven with an ideal current source. The mouse was modeled by importing a copy of a freely available model (Digimouse[48]), where the tissue properties of the mouse atlas-map were assigned according to the operating frequency of the electromagnetic fields[77]. The set-up of our 1-photon Ca$^{2+}$ imaging experiments (Fig. 1d) was replicated. A uniform mesh grid resolution of 5 mm x 5 mm x 5 mm was applied.

**Reporting summary**. Further information on research design is available in the Nature Research Reporting Summary linked to this article.

## Data availability

The data that support the main findings of this study are publicly available in the Buzsaki Lab Databank: https://buzsakilab.nyumc.org/datasets/YaghmazadehO/.

## Code availability

MATLAB script packages used in the analysis of this study are downloaded from https://github.com/MouseLand/Kilosort, and https://github.com/ettierguillaume/MiniscopeAnalysis, and custom scripts specific to this paper can be found at https://github.com/omidyaghmazadeh.

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

## Acknowledgements

Authors thank Bei Zhang for her help in the beginning of this project on conducting preliminary simulations. Authors thank all the members of the BuzsakiLab (buzsaki-lab.com) for their support and comments on different aspects of the work in progress. O.Y. thanks Manuel Valero, Peter Petersen, Victor Varga, Yiyao Zhang, Yuta Senzai, Eliezyer Fermino de Oliviera and Ipshita Zutshi for their help with different aspects related to the experiments. This work was supported by the following grants: NIH-R01 (# 1R01NS113782-01A1) and TL1 postdoctoral fellowship (# 2TL1TR001447-06A1) to OY.

## Authors contributions

G.B. and O.Y. conceived the project with support from L.A. and D.K.S. for the RF aspects. O.Y., G.B, and M.V designed the experiments. O.Y. and M.V. performed the experiments and associated data analysis. Simulations were conceived by CC, GC, and OY and performed by GC. O.Y. and G.B. wrote the paper and all authors participated in its revision.

## Competing interests

The authors declare no competing interests.

## Ethics statement

All experiments were conducted in accordance with the Institutional Animal Care and Use Committee (IACUC) of New York University Medical Center.
