## [Peer Review File · Communications Engineering]

nature portfolio

Peer Review File

Neuronal activity under transcranial radio-frequency stimulation in metal-free rodent brains in-vivoPeer Review Information

Journal: Communications Engineering

Manuscript Title: Neuronal activity under transcranial radio-frequency stimulation in metal-free rodent brains in-vivo

Corresponding author name(s): Omid Yaghmazadeh & György Buzsáki

Editorial Notes:

Transferred manuscripts This manuscript has been previously reviewed at another *Nature Research* journal. This document only contains reviewer comments and letters for versions considered at *Communications Engineering*.

Reviewer Comments & Decisions:

We thank the Reviewers for their constructive suggestions. We have revised our manuscript according their critique and comments. The main changes are highlighted in the revised version of the manuscript.

Reviewers' comments:

Reviewer #1 (Remarks to the Author):

Yaghmazadeh et al. investigated the effect of radiofrequency (RF) stimulation on neuronal activities in rodents. They first show that the spike rates of a significant fraction of neurons recorded using metal electrodes are affected by RF stimulation. However, by carefully excluding RF-interference with recording systems, the authors show that RF stimulation did not change neuronal activities detected with calcium imaging. These data suggest that RF stimulation will interact with metal parts in the recording system to cause notable changes in the spike frequency of some neurons. Because the level of RF strength in this study is much higher than that of our daily exposure, RF exposure in our daily lives would not affect ordinary people's brain activities.

Overall, the experiments are well performed, and the data are appropriately analyzed and clearly displayed in figures. The results are carefully discussed, and the conclusion is convincing. As far as I noticed, all previous work investigating RF stimulation's effect on neuronal activities did not use RF-interference-free optical imaging. Therefore, this piece of work done by physiology experts would become an "instant classic" among papers of a similar kind. Nevertheless, I am only concerned that this study does not make any new conceptual advances in biology, which would be essential to be accepted for publication in *Communications Biology*.

We thank the reviewer for highlighting the positive aspects of our study.

Minor comments:

1. Suppl. Fig. 1 is about temperature change upon RF stimulation, which is not matched with the description in the text: “Our temperature measurements showed that the head-cap, including the copper mesh, was not blocking the penetration of RF into the soft tissue, skull and brain from both sides of the head.” To show the effect of the head cap on RF penetration, the authors have to compare RF signals (temperature changes here) with and without it. Correct the statement.

In response to this comment, we have performed the needed experiment. We now illustrate that the added copper mesh shielding does not prevent the RF fields to penetrate into the brain (new Suppl. Fig. 1j).

2. line 223: this is a confusing statement with $p < 0.05$ for no significant difference. This part seems to refer to Fig. 3h and should cite $p = 0.46$ instead.

We thank the Reviewer for catching this oversight. Error is corrected.

3. Suppl. Fig. 9b-d: The signal is normalized, and it would be difficult to assess the signal distribution within the brain because of the prominent peak outside the brain. It might be more informative if additional panels show signal distribution only within the brain (only inside the dashed circles).

We thank the Reviewer for this useful suggestion. We added those panels accordingly.

4. Owen et al. (Nature Neuroscience 22:1061–1065, 2019) show that a slight change in tissue temperature can affect neuronal firing. It also provides an insight into the time course of the effect. This paper (possibly together with Christie et al., Neuroimage 66: 634–641, 2013) might be better to be cited in the discussion on temperature effects on neuronal firing (line 389).

We thank the Reviewer alerting us to these relevant references. Both references are cited as suggested. The first paper was already among our list of references and the second one is added.

Reviewer #2 (Remarks to the Author):

This is an interesting paper studying effects of RF radiation on neural activity. The results suggest that RF did not modulate neural activity when the neural activity was monitored using optical imaging. On the other hand, changes in neural firing rates were observed when neural activity was recorded using metal containing Si probes. These findings are really intriguing. However, there are some major questions and concerns:

We thank the Reviewer for the favorable evaluation of our work as well as providing useful feedback to us.

1) It is reported in the paper that to quantify the responses to RF stimulation, the authors used the area-under-the-curve of the Ca²⁺ transient traces and the deconvolved spiking activity derived from it. Considering Ca is a secondary measure of spiking activity and the reports from literature pointing into inconsistencies between calcium fluorescence change and electrophysiological recordings, I am not sure how one can be very certain in comparing results from two methodologies. It is highly possible that small but significant changes in firing frequency may not be translated into any changes in calcium. Authors need to address this discrepancy and potential errors due to that in the paper, including the abstract and discussions.

We cannot agree more. It has been shown repeatedly that changes in Ca²⁺ transients do not follow an exact match with the neuron's sodium spiking activity. Even in in-vitro preparation, single action potentials are rarely resolved. Instead, Ca²⁺ transients correlate much better with spike bursts, which are known to involve Ca²⁺ influx. Yet, Ca²⁺ imaging (whether 1-photon or 2-photon) methods are still considered a reliable measure of neuronal activity and constitute the state-of-the-art for optical recording of neural activity. The term "neuronal activity" is typically vaguely defined and can vary from spikes in pyramidal cells and inhibitory interneurons, EPSCs, IPSCs, Ca²⁺ spikes, plateau potentials, etc. Thus, Ca²⁺ imaging may be considered as a method more sensitive to measure "neuronal activity" of various kinds, not only spiking. We make this distinction clearer in our revised manuscript.

2) Authors suggest that but does not provide a mechanistic explanation on the observed neural response change in the case of metal electrodes. Electromagnetically induced current loops might be formed in the metal wires or between the metal wires and the tissue, however, the only overall effect one would expect in this frequency is local heating due the resistive nature of metals. I think it very important to measure local temperature and heating for the case of metal-based probes and have some clarification on this possibility. There are various methods to measure the temperature in this configuration.

We thank the Reviewer for this excellent suggestion. We have performed a new experiment that illustrates that the presence of metal electrodes in the brain induces a large temperature increase which signifies a local enhancement of the induced electric fields. For this experiment, we used a triplet of 50µm diameter tungsten wires and a small-footprint (300µm diameter) optical temperature sensor and measured brain temperature change in the brain both in the presence and absence of metal wires. (shown in new Suppl. Fig. 9).

3) It looks like electrophysiological recording experiments were performed in 3 mice and 2 rats. Did the authors observe a difference between these two species?

Separating the recorded cells between the two species, we did not observe a qualitative difference: a considerable fraction of neurons in both species are affected by RF exposure with statistical significance (33.1% in mice and 22.3% in rats). Given the smaller head, thinner skull and thinner soft tissue surrounding the head of the mouse, we only assume that the same RF power might exert a stronger quantitative effect in the mouse compared with the rat.

Unfortunately, we do not have large numbers of animals to make this statement reliable as the variability between animals of the same species can be as large as across species. Our emphasis is on the qualitative similarity. To respond to the Reviewer with the data at our disposal, in Fig. 2d we show neurons from mice and rats with different symbols to highlight this similarity.